# Assessment of Direct Economic Losses of Flood Disasters Based on Spatial Valuation of Landuse and Quantification of Vulnerabilities: A Case Study on the 2014 Flood in Lishui city of China

**Haixia Zhang[1, 2], Weihua Fang[1, 2, 3], Hua Zhang[1, 2], Lu Yu[1, 2]**

[1] Academy of Disaster Reduction and Emergency Management, Ministry of Emergency Management and Ministry of Education, Beijing Normal University, Beijing 100875, China

[2] Key Laboratory of Environmental Change and Natural Disaster of Ministry of Education, Faculty of Geographical Science, Beijing Normal University, Beijing 100875, China

[3] Southern Marine Science and Engineering Guangdong Laboratory, Guangzhou 511458, China

**Correspondence:** Weihua Fang (weihua.fang@bnu.edu.cn)

**Abstract.** Detailed and reliable assessment of direct economic losses of flood disasters is important for emergency dispatch and risk management in small and medium-sized cities. In this study, a single flood disaster in Lishui city in 2014 was taken as an example to study and verify a method for rapid and detailed assessment of direct economic loss. First, based on a field investigation, the inundation data simulated by the one-dimensional hydrodynamic model and geographic information system (GIS) analysis method were verified. Next, the urban landuse map and high-resolution landuse classifications based on remote sensing data were fused and combined with expert questionnaire surveys, thereby providing the 47 types and values of landuse. Then, based on the previous depth-damage function in the past study, the vulnerability curves of 47 types of landuse in Liandu district were fitted by the lognormal cumulative distribution function and then calibrated using disaster loss report data. Finally, the distributions of the loss ratio and loss value were estimated by spatial analysis. It is found that the landuse data has detailed types and value attributes as well as high resolution. Secondly, the vulnerability curves after function fitting and calibration effectively reflect the change characteristics of landuse loss ratio in this area. Finally, except for the 3 types of land for agriculture, recreational and sports facilities, and green parking spaces, the optimized simulated total loss is 322.6 million RMB, which is 0.16% higher than the statistics report data. The estimated loss ratio and loss value can reliably reflect the distribution pattern of disaster losses in detail, which can be applied by government and private sectors to implement effective disaster reduction and relief measures.

**Key words:** Flood disaster, Direct economic loss, Loss assessment, Vulnerability curve

## 1 Introduction

Detailed and reliable assessments of the direct economic losses of flood disasters are very important in disaster emergency rescue and urban flood risk management (Li et al., 2017; UNISDR, 2015). The results of a rapid quantitative assessment of disaster losses with high spatial resolution not only provide suggestions for the government to formulate emergency dispatch

management measures, such as releasing disaster information, deploying rescue forces and relief materials, and the emergency resettlement of disaster victims, but also lay a data foundation for decision-makers to plan sponge cities and formulate flood risk management systems and climate change adaptation policies (Alfieri et al., 2016; Merz et al., 2010). Among them, sponge city refers to a city that can be as flexible as a sponge in adapting to environmental changes and natural disasters. It absorbs, stores, infiltrates and purifies water when it rains, and releases and reuses the stored water when necessary (Yu et al., 2015). In order to solve the problems of water shortage, waterlogging and water pollution caused by rapid urbanization, the Chinese government launched the sponge city construction plan on December 31, 2014 (MHURD, 2014), which can effectively alleviate urban waterlogging, reduce runoff pollution, save water resources and improve ecological environment.

As a key component of flood risk assessment, flood loss assessment has been extensively analyzed by researchers (Falter et al., 2015; Koks et al., 2015). The flood loss data obtained from a comprehensive high-quality field survey after a disaster can accurately reflect the disaster loss situation and has important reference value for the establishment and verification of flood loss models (Carisi et al., 2018). However, given that loss data can only be obtained after a flood, these data cannot provide timely guidance for disaster relief. In addition, the collection of the loss data is also time-consuming and laborious.

Due to the development of existing flood loss assessment models, there are relatively mature methods and tools (EMA, 2002; Scawthorn et al., 2006), and the popularization of flood insurance provides relatively complete socioeconomic and disaster loss data, thus, disaster losses can be quickly assessed when floods occur (Hsu et al., 2011). The United States (Custer and Nishijima, 2015; USACE, 2006), the United Kingdom (Stephenson and D'Ayala, 2013), Japan (Dutta et al., 2003), Canada (NRC, 2017), Australia (Hasanzadeh Nafari et al., 2016b, 2016a; Wehner et al., 2017), Italy (Amadio et al., 2016), China (Li et al., 2012; Penning-Rowsell et al., 2013), and other flood-prone countries have carried out a large number of loss assessment studies using different classification systems of exposure data and then used the existing loss database and post-disaster investigation data to establish local flood vulnerability functions.

In addition, with the development and application of geographic information systems (GISs), remote sensing (RS), hydrological models, and hydrodynamic models (Elkhrachy, 2015; Jonkman et al., 2008), flood loss assessment models based on depth-damage functions have been improved (Komolafe et al., 2018). However, there are still some problems. First, there is a lack of a depth-damage functions for use in specific areas, which need to be constructed through extensive post-disaster survey data (Albano et al., 2018). Second, the effect and accuracy of the assessment are affected by the scale of the exposure data. The microscale loss assessment model for each affected object (building, infrastructure object, etc.) has poor applicability. However, mesoscale exposure data mainly refer to landuse obtained through remote sensing (RS) interpretation (Merz et al., 2010). Although mesoscale data can effectively be used to extract the spatial distribution of buildings, it is difficult to identify the occupancy types of buildings. These problems lead to high uncertainties and disparities in flood loss assessments (Gerl et al., 2016; Pinelli et al., 2020).

With the introduction of fuzzy mathematics, grey system models, genetic algorithms, and other mathematical methods, the rapid estimation and prediction of regional flood direct economic loss can be realized (Qie and Rong, 2017; Zhao et al., 2014; Zhou et al., 2006), which can effectively reflect the overall situation of economic losses in a large region and reduce the

65 investment in human and material resources. However, due to the lack of high-resolution location information, this approach cannot provide timely and effective suggestions for the government to formulate targeted emergency scheduling plans.

To effectively improve the accuracy, timeliness, and practicability of flood disaster loss assessment, a refined assessment model of single-flood disaster losses in small- and medium-sized cities was explored in this paper. The heavy rainfall from 18 August 2014 to 20 August 2014 caused severe river backflow and urban waterlogging, and many houses and roads were

70 flooded in Lishui city. Therefore, taking this flood disaster as an example, a refined assessment model for the direct economic losses of regional flooding disasters was developed. The model included flood inundation simulation, landuse type fusion, landuse value quantification, vulnerability function fitting and optimization, as well as loss ratio and loss value estimation.

## 2 Materials

### 2.1 Study area

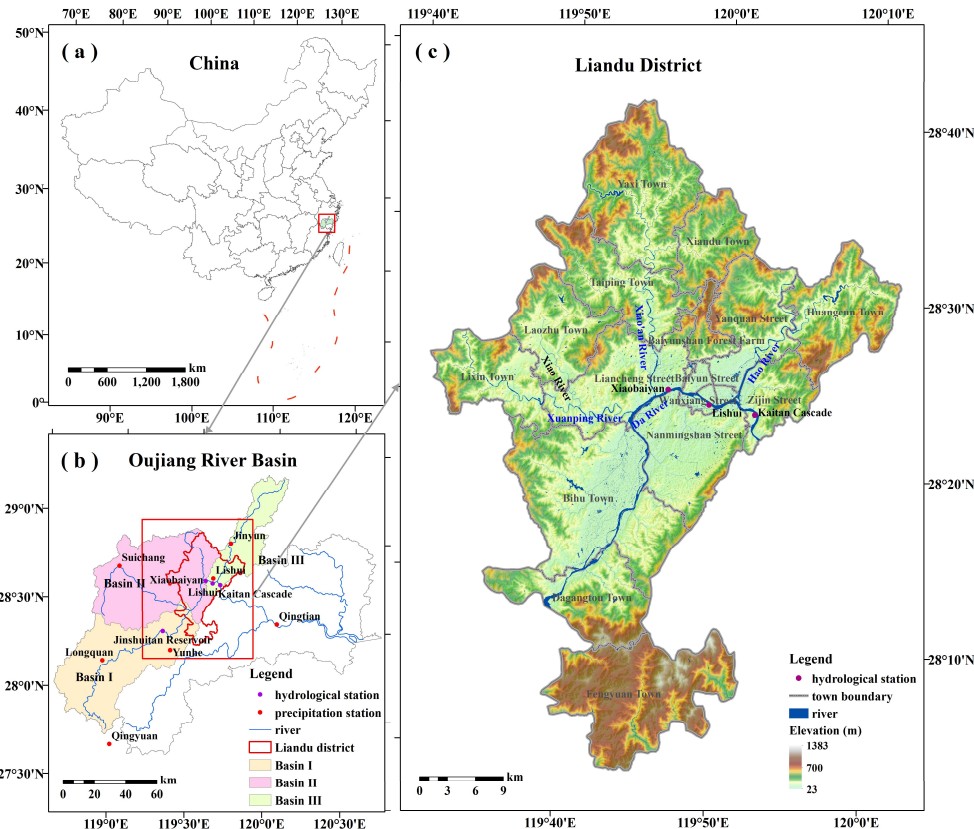

**Figure 1. Location of the study area. (a. The location of Oujiang River Basin in China. b. The distribution of hydrological stations, precipitation stations, rivers and sub basins in Oujiang River Basin, and the location of Liandu district in Oujiang River Basin. c. The terrain distribution and town boundary of Liandu district.)**

Liandu district is in southwestern Zhejiang Province, between 28°06′ N-28°44′ N and 119°32′ E-120°08′ E (Figure 1). This district is in the middle reaches of the Oujiang River Basin, surrounded mountains and a plain in the middle, with a total area of approximately 1,502 km$^2$. Liandu is a district under the jurisdiction of Lishui city, with a relatively concentrated population and socioeconomic status. As a result of both urban planning and topography, the flood disaster in Liandu district causes heavy losses. Because topography has a great impact on hydrology and hydrodynamics, it is easy to ignore regional differences based on administrative units. Considering the impact of the Jinshuitan Reservoir operation, the upper reaches of the Oujiang River Basin are divided into three sub watersheds (Figure 1b).

## 2.2 Precipitation

The gridded precipitation data come from the hourly precipitation data set of the National Meteorological Information Center, which integrates China's automatic station data with the NOAA CDR Climate Prediction Center morphing technique (CMORPH) product with a resolution of 0.1°. The overall error is within 10%, and the accuracy in areas with heavy rainfall and sparse sites is greater than in similar international products (Shen et al., 2014). The data can effectively reproduce the spatiotemporal pattern of rainfall and are suitable for simulating flood inundation.

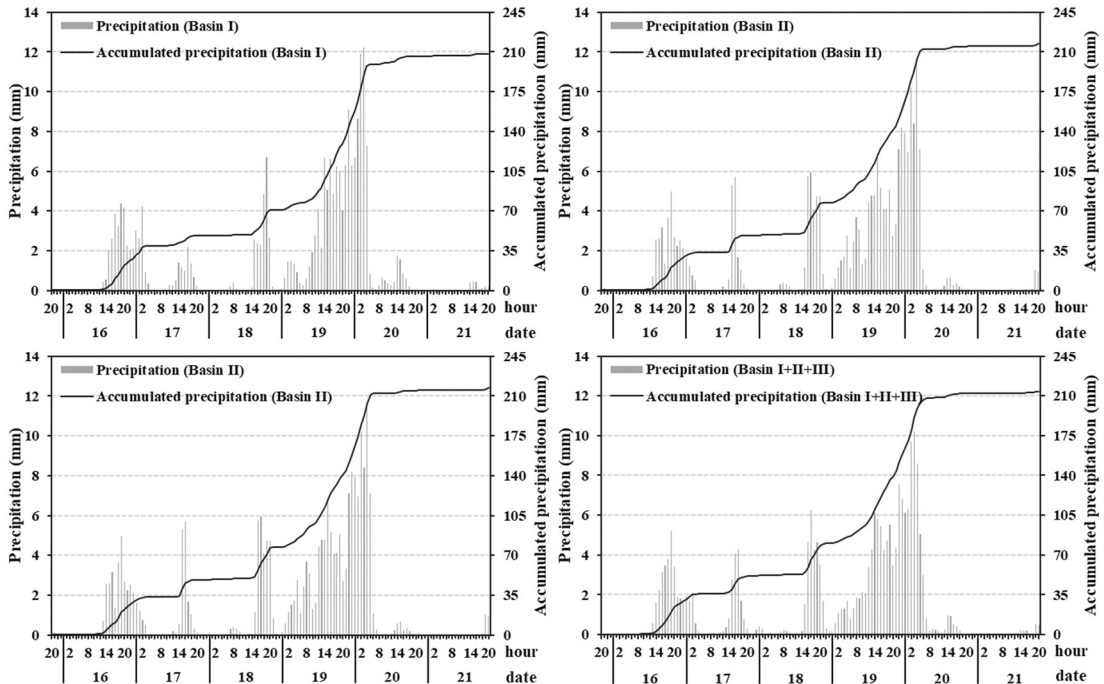

**Figure 2. Distribution of hourly and accumulated precipitation in the upper reached of the Oujiang River Basin (Basin I + II + III) and each sub basin. (The grey column is hourly precipitation, and the black curve is accumulated precipitation. The precipitation time is from 20:00 on August 15th to 20:00 on August 21st, 2014.)**

Based on the hourly precipitation levels, the mean accumulated precipitation of the upper reaches of the Oujiang River Basin and the 3 sub watersheds were calculated (Figure 2). The precipitation change trend of each sub watershed was generally the

same, with the accumulated precipitation exceeding 210 mm. The precipitation increased rapidly after 5:00 on August 18th, and the entire precipitation process basically ended at 10:00 on August 20th.

## 2.3 Water level and flood inundation

The water level data come from the hourly observations of hydrological stations of the Zhejiang Water Resources Department, including the measured water level, warning water level, and guaranteed water level. Xiaobaiyan and Lishui are river stations, and the Jinshuitan Reservoir and the Kaitan Reservoir are reservoir stations (Figure 1). Based on the hourly measured water level, beginning at 12:00 on August 19th, the water levels of the Jinshuitan Reservoir, Xiaobaiyan, and Lishui increased significantly, while the water level of the Kaitan Reservoir first dropped slightly and then increased significantly, and the water levels at these stations reached a peak at approximately 12:00 on August 20th. The water levels at Xiaobaiyan, Lishui, and the Kaitan Reservoir returned to normal at 00:00 on August 21st, while that at the Jinshuitan Reservoir dropped to a certain water level at 00:00 on August 22nd, and the subsequent downward trend was slow (Figure 3).

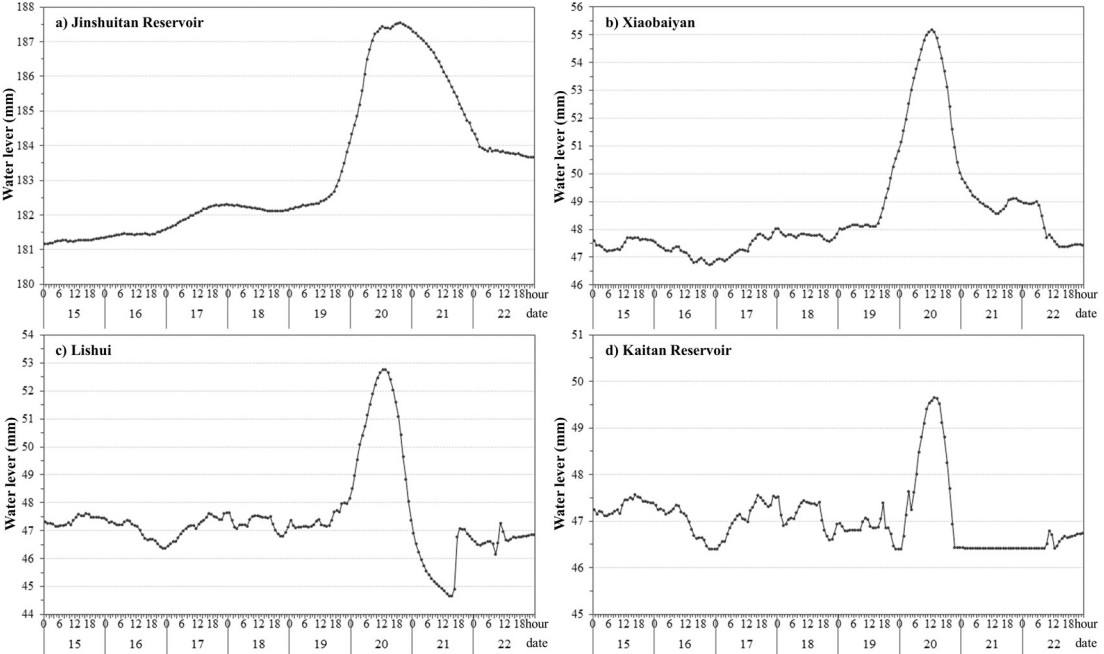

**Figure 3. Distribution of hourly water levels at hydrological stations. (The time of water level data is from 00:00 on August 15th to 00:00 on August 23rd, 2014)**

The flood inundation was calculated from the Yuxi to Kaitan Reservoir hydraulic model constructed by *Zhejiang Design Institute of Water Conservancy & Hydro-Electric Power*. In this study, the middle and upper reaches of the Oujiang River Basin were generalized into a mathematical model, the unsteady flow partial differential equations of the Saint-Venant open channel were used to describe the flood evolution process, and the implicit difference method was used to transform it into a difference equation. Newton iteration and Gaussian principal component elimination method were used to solve the problem

time by time, so as to obtain the water level and discharge of each section (Kang and Chen, 2007). This model considered weirs, gates, water-blocking bridges, water exchange between intervals, and flood detention areas, etc., and can be applied to the quantitative analysis and calculation of the flood evolution of this section of the river.

In order to make the flood evolution calculation can better simulate the water depth of this basin, the measured river section and the flood in 2014 were selected for simulation calculation to verify the accuracy of the mathematical model of the flood evolution calculation, and determine the parameters of the model. The simulation result of the measured flood on August 20, 2014 was carried out. The upper boundary used the discharge process of each reservoir and the measured data from the hydrological station, and the lower boundary used the measured water level. Comparing the model result with the measured
flood traces and the measured process at the hydrological station, the difference in water level between the two was 0 m - 0.09 m, which shows that the model parameters were reasonable. The flood volume was calculated based on the simulated water level and the elevation of the embankment (Table 1). Based on the overflow volume and topographic data, the submerged area and inundation depth were estimated by GIS tools. Through measured flood traces, field surveys and aerial photograph, it was found that the simulated submergence results can well reflect the actual flood.

**Table 1. Overflow volume of Liandu district in 2014 based on the hydraulic model (10,000 m³)**

|  | Da River South | Da River East | Da River North (Da River Section) | Da River North (Hao River Section) |
|---|---|---|---|---|
| **Overflow volume** | 230.63 | 264.81 | 60.17 | 277.51 |

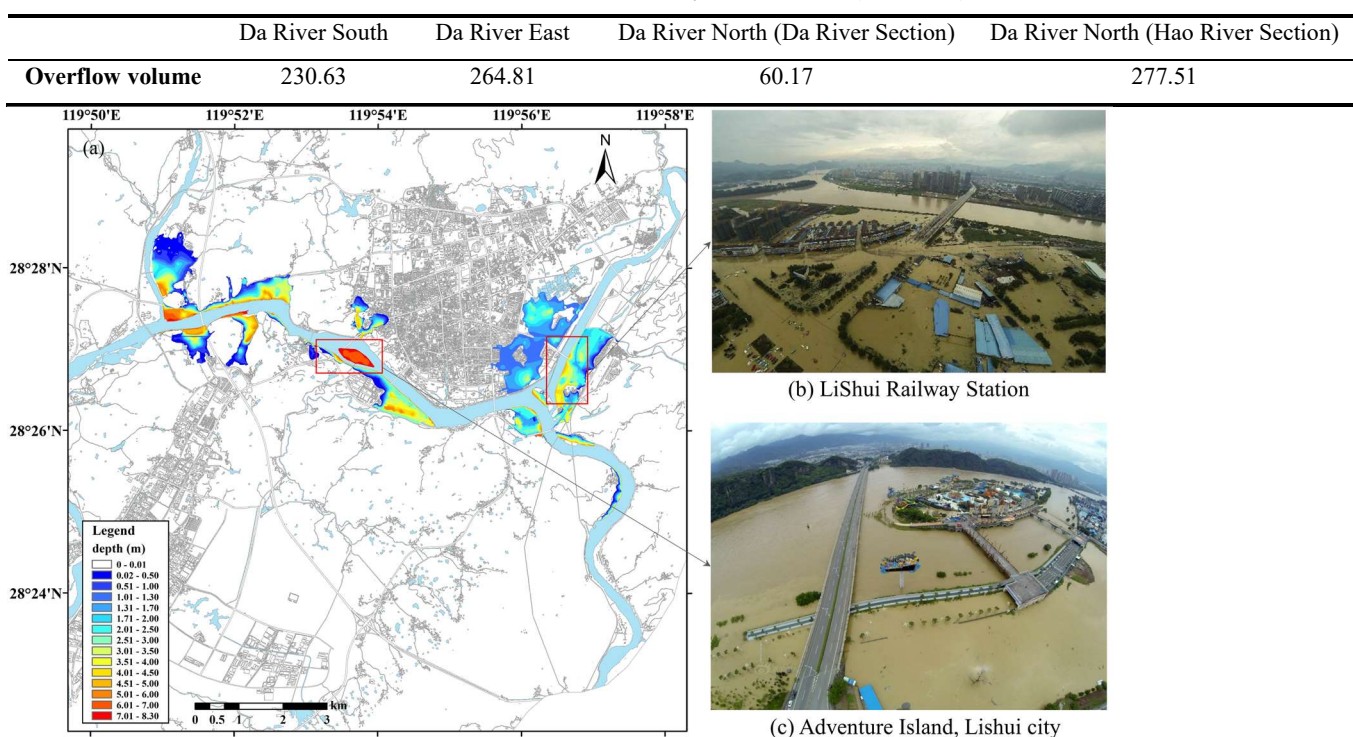

**Figure 4. (a) Distribution of maximum simulated submergence depth, (b, c) Aerial photographs of Liandu district in 2014.**

Based on the simulated maximum inundation depth distribution (Figure 4a) and real-time aerial photographs (Figure 4b, Figure 4c), the flood inundation area was mainly concentrated in the river confluence and both sides of the river, the inundation

depth decreased from the river bank to both sides, and the inundation depth of the central island was generally greater than 7 m.

## 2.4 Disaster loss reporting

The disaster loss report was obtained from the Lishui Civil Affairs Bureau, which was reported by the local government. The report recorded 41 statistical indicators, including the affected population, affected area of crops, agricultural losses, infrastructure losses, public welfare facility losses, household property losses, and direct economic losses. According to these statistics, a total of 167,300 people was affected in Liandu, 17,330 people were relocated in emergencies, 247 rural houses collapsed, and the direct economic loss was approximately 377.15 million RMB.

The insurance claim dataset came from the auto insurance list of the catastrophe "Zhejiang 0819 Rainstorm" of the Lishui branch of the People's Insurance Company of China (PICC), and the dataset recorded 19 indicators, including the policy number, the information of the insured, the estimated compensation, and the compensation paid. As of August 24th, a total of 1,045 motor vehicle insurance reports were received, with a reported loss of 50.7969 million RMB and a decided compensation of 50.6893 million RMB. According to the analysis of the market share of various insurance types in Zhejiang Province in December 2014, PICC motor vehicle insurance accounted for approximately 48.357% of Lishui city.

## 3 Methods

In this study, an assessment model of the direct economic loss ratio and loss value of flood disasters was developed by utilizing methods such as landuse type fusion, landuse value estimation, vulnerability curve fitting and optimization.

## 3.1 Data fusion of landuse types

The distribution of landuse types was obtained through the fusion of current landuse data in Lishui city with high resolution landuse classification results based on remote sensing data. The former data came from the urban and rural space development current status map of the Natural Resources and Planning Bureau in 2013, which was divided into 47 categories (Table 2) according to the *Code for classification of urban landuse and planning standards of development land* (MHURD, 2011). The landuse classification results were derived from the Gaode map with a resolution of 2.3870768 m, including 5 categories: transportation, grassland, waters, agriculture and forest, and buildings. The fusion steps of vector data and raster data are as follows.

1) First, we selected the unified geographic coordinate system used in this study. In order to be consistent with the flood inundation results, we interpolated the landuse classification results into raster data with a resolution of 2 m by the nearest neighbour interpolation method. Then the vectorized current landuse data was spatial adjusted to make it overlap with the position of the landuse classification results.

2) Second, we extracted the landuse type in the current landuse data corresponding to the location of each building pixel in the landuse classification result, and assigned the type to the corresponding building pixel. The building pixels that have not been reassigned were assigned according to the adjacent building types. In addition, the road within 2 m of the residence was set as community parking, and the buildings far from urban areas were set as the village construction land.

3) The water, agriculture and forest, and road in the landuse classification results were assigned as water, agriculture and forest, and urban road, respectively. The agriculture and forest in areas with urban buildings and the agriculture and forest and grassland in park areas were assigned as park green land.

4) Whether there were un-reassigned pixels in the landuse classification results. If so, these pixels were reclassified according to the corresponding current land type.

## 3.2 Estimation of landuse values

The value of landuse obtained from expert questionnaires is relatively reliable, and the steps are as follows.

1) Based on the *Lishui City Master Plan* (2013-2030), *Lishui City 13th Five-Year Plan*, and *Lishui City Statistical Yearbook in 2015*, we gave the reference information of current area, planned area, planned investment, unit area budget, and description of landuse types.

2) Due to the different disaster characteristics of landuse, the four major categories of residential, commercial, public management and public services, and industrial were estimated the exposure value of the indoor properties, and the other categories were estimated the value or cost per unit area, as shown in Table 2.

3) We issued questionnaires to 7 experts in fields such as municipal engineering design, construction industry, water design, ecological city planning, and natural disasters, and invited experts to estimate the landuse value based on their professional background knowledge and the actual situation of the study area.

4) The value of each landuse was determined by collating the questionnaires and calculating the average values.

## 3.3 Calibration of vulnerability curves

Although the vulnerability curves of different regions are different, there are similarities in the trend of the loss ratio with the inundation depth, and we can learn from them. Therefore, based on the existing vulnerability curves in many countries and regions (Coto, 2002; FEMA, 2013; Mo and Fang, 2016; Shi, 2010), the fitting and optimization steps of vulnerability function are as follows.

1) According to the existing depth-damage database, the relationship between inundation depth and loss ratio of each landuse in Liandu district was developed. Since the HAZUS-FLOOD had a complete building occupancy class, this study mainly referred to it. First, we developed the comparison table between the building occupancy class in HAZUS-FLOOD and the landuse type in Liandu district. Second, based on the inundation depth in Liandu district and the water depth in HAZUS-FLOOD, the range of inundation depth in the depth-damage function was set and the unit of water depth was converted to meters. Finally, the HAZUS-FLOOD was summarized according to the building occupation class, and the average loss ratio

of all samples for each building occupancy class under each inundation depth was calculated, which was used as the reference of the loss ratio of corresponding landuse types under the same inundation depth in Liandu district. If there was no similar building occupancy type in HAZUS-FLOOD, other databases were referenced (Figure 6).

2) The appropriate function was selected to fit the curve of inundation depth and loss ratio for each landuse type developed in step 1. In the previous study, the vulnerability curve can be fitted by a polynomial, a power function (Büchele et al., 2006), or logistic regression (Cao et al., 2016), and it can also be smoothed by nonparametric forms such as the kernel density (Merz et al., 2004). In this paper, the lognormal cumulative distribution function (Limpert et al., 2001) was selected to fit the vulnerability curve. The formula is as follows:

$$y = F(x, scale, shape) = F(x|\mu, \sigma) = \frac{1}{\sigma\sqrt{2\pi}} \int_0^x \frac{1}{t} e^{\frac{-(\log t - \mu)^2}{2\sigma^2}} dt, \quad x > 0 \tag{1}$$

where $y$ is the loss ratio, $x$ is the inundation depth, $\sigma$ and $\mu$ are the standard deviation and mean of the $\log x$, respectively. For lognormal cumulative distribution function $F$, the shape parameter $shape = \sigma$, which affects the shape of the distribution, and scale parameter $scale = e^{\mu}$, which affects the stretching and shrinking of the distribution.

3) Based on the vulnerability function fitted in step 2, the loss ratio and loss value of landuse in Liandu district were estimated. First, the two raster layers of landuse type and inundation depth were overlaid. Then, the loss ratio of each grid was calculated based on the vulnerability function. If the inundation depth of the grid was 0 m, the loss ratio was also 0. Otherwise, the corresponding vulnerability function was searched according to the landuse type of the grid, and its loss ratio was calculated based on the inundation depth of the grid. Finally, the loss value of each grid was calculated by Eq. 2.

$$L = DR * V \tag{2}$$

where $L$ is the loss value of the landuse, $DR$ is the loss ratio of the landuse, and $V$ is the value of the landuse.

4) The vulnerability function was optimized by disaster loss reporting data. First, the mapping table between the statistical indicators of disaster loss reporting and landuse types in Liandu district was developed (Table 2). Second, the simulated total loss of landuse in Liandu district corresponding to the loss reporting data of indicator $k$ was calculated by Eq. 3. Then, the non-linear equation was established with the minimum error between the disaster reporting loss and the simulated total loss as the objective function, and the scaling factor $a_k$ was solved by the least square method.

$$TL_k = \sum_{i=1}^{n_k} \sum_{j=1}^{m_{ik}} L_{ij} = \sum_{i=1}^{n_k} \sum_{j=1}^{m_{ik}} F(x_{ij}, shape_i, a_k * scale_i) * value_i \quad (k = 1,2,...,5) \tag{3}$$

where $TL_k$ is the simulated total loss, $n_k$ is the number of landuse types corresponding to the disaster loss reporting indicator $k$, $m_{ik}$ is the number of grids of the landuse type $i$ corresponding to the indicator $k$, $L_{ij}$ and $F(x_{ij}, shape_i, a_k * scale_i)$ are the loss value and loss ratio of the grid $j$ of the landuse type $i$, respectively. $value_i$ is the asset value or cost per unit area of landuse type $i$, $x_{ij}$ is the inundation depth of the grid $j$ of the landuse type $i$, $shape_i$ and $scale_i$ are the parameter of the vulnerability function of the landuse type $i$, $a_k$ is the scaling factor of the indicator $k$, and the initial value is 1.

5) Based on the optimized vulnerability function, we used the method in step 3 to re-estimate the loss ratio and loss value of landuse in Liandu district.

**4 Results and Analysis**

**4.1 Distribution of landuse types**

The distribution of high-resolution landuse types in Liandu was obtained by using the data fusion method (Figure 5a), and the names and codes of landuse types were shown in Table 2. The fused landuse type data effectively integrate the respective advantages of the current urban landuse and landuse classification results based on remote sensing images. The data not only have high-resolution spatial location information but also reflect the detailed types of landuse.

    Agricultural and forestry land in Liandu district is the most widely distributed type. Woodlands are mainly distributed in the
hilly areas of the north, east, south, and northeast. The built-up area of Liandu district is distributed along the river in a block shape, among which residential land is mainly distributed in communities near the river. Industrial land is mainly distributed in north-eastern Wanxiang Street and the Economic and Technological Development Zone, which is currently in the development stage, and many industrial plants have been built.

    To strengthen intraregional connections, the roads and traffic facilities are relatively complete, with an urban road area of
4.33 km$^2$. Commercial, warehouse, public management, and public service facilities are relatively small and scattered. They are mainly concentrated near residential and industrial land, providing various services. Park green space is distributed along the river or close to residential and commercial land, while square green space, protective green space, and public facilities are small and scattered.

**4.2 Distribution of landuse values**

The asset value or cost per unit area of landuse types was estimated based on expert questionnaires (Table 2). Commercial and industrial land has many internal equipment and items with the highest asset unit price. Public management and public service facilities and residential land have a higher indoor property value. The value of agricultural and forestry land is the lowest.

    The spatial distribution of landuse value in Liandu in 2013 was calculated through grid assignment (Figure 5b). The value of assets per unit area in the northern city and Wanxiang Street is generally high due to the concentration of commercial,
industrial, residential, public management, and public service facilities in the area. There is a water amusement project on the central island, which has a higher value per unit area. Many industrial plants are distributed in the Economic and Technological Development Zone, but due to the development stage and incomplete internal facilities, the unit price of the industrial land in this zone is calculated at half of the estimated value. Agricultural and forestry land is widely distributed and low in value, so most areas of Liandu have low values.

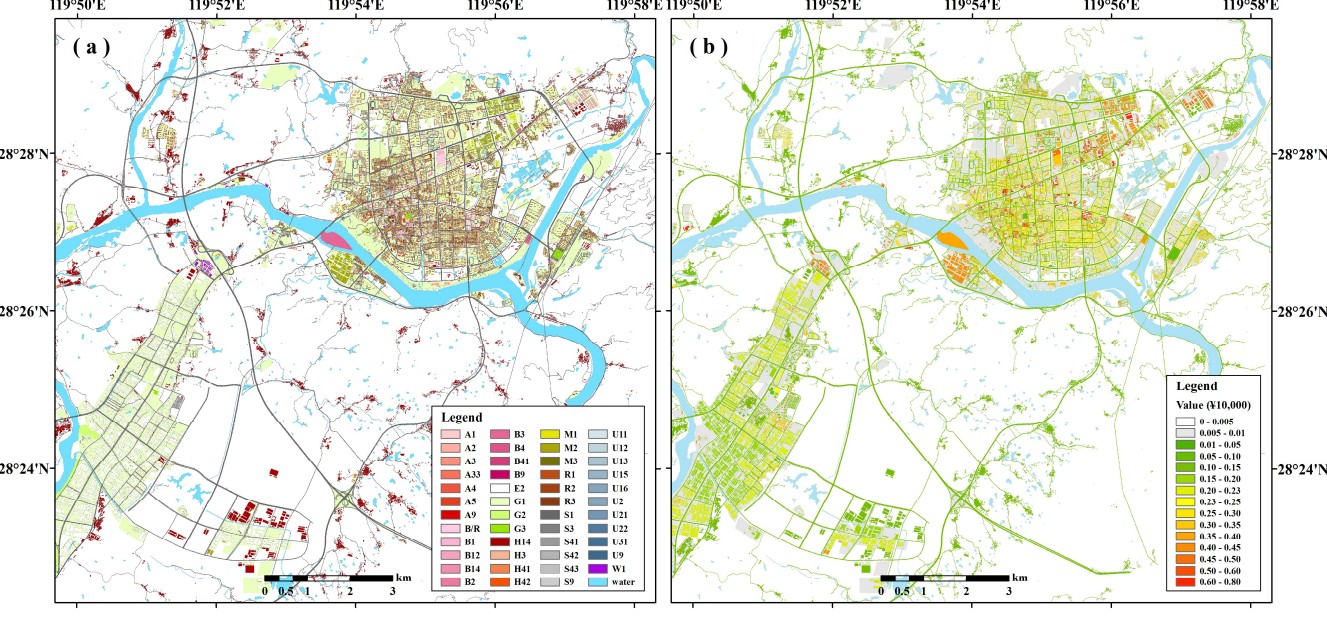

**Figure 5. Distribution of landuse types (a) and landuse values (b) in the central urban area of Liandu district in 2013.**

### 4.3 Fitted vulnerability curves

The vulnerability curves of all landuse types were fitted by a lognormal cumulative distribution function based on the matrix of the inundation depth and loss ratio. Based on the comparison of simulated losses and disaster loss report data (Table 2), we optimized the scale parameters of the vulnerability function (Figure 6) through the vulnerability curve calibration method (Section 3.3).

**Table 2 The Classification, value, inundated area of landuse types, and the comparison of simulated loss before optimization, simulated loss after optimization and statistics loss of each landuse type in Liandu district.**

| Code | Classification | Value (¥10,000/4 m$^2$) | Total value (¥10,000) | Inundated area (m$^2$) | Mean loss ratio of inundated area | Simulation loss before optimization (¥10,000) | Simulation loss after optimization (¥10,000) | Statistical loss (¥10,000) |
|---|---|---|---|---|---|---|---|---|
| A1[*] | administrative office | 0.40 | 13,892 | 2,020 | 0.79 | 160 | 131 | |
| A2[*] | cultural facilities | 0.35 | 3,310 | 24 | 0.81 | 1.7 | 1.1 | |
| A3[*] | educational and scientific research | 0.25 | 13,914 | 15,980 | 0.64 | 639 | 500 | 741 (public welfare facilities) |
| A33[*] | primary and secondary schools | 0.20 | 5,947 | 7,296 | 0.31 | 112 | 77 | |
| A4[*] | sports land | 0.30 | 1,311 | 0 | 0.00 | 0 | 0 | |
| A5[*] | medical and health land | 0.35 | 4,601 | 568 | 0.80 | 40 | 32 | |
| A9[*] | religious land | 0.10 | 289 | 0 | 0.00 | 0 | 0 | |
| B/R[*] | commercial and residential land | 0.35 | 8,416 | 144 | 0.45 | 5.7 | 1.9 | 2940 |

| | | | | | | | | |
|---|---|---|---|---|---|---|---|---|
| B1[*] | commercial facilities | 0.46 | 9,945 | 5,908 | 0.89 | 603 | 473 | (industry and commerce) |
| B12[*] | wholesale market land | 0.50 | 15,358 | 16,252 | 0.48 | 982 | 458 | |
| B14[*] | hotel land | 0.30 | 3,248 | 2,520 | 0.29 | 55 | 19 | |
| B2[*] | business facilities | 0.60 | 11,072 | 1,100 | 0.10 | 16 | 2 | |
| B4[*] | business outlets of public facilities | 0.35 | 19 | 148 | 0.47 | 6 | 3 | |
| B41[*] | gas station | 0.44 | 250 | 0 | 0.00 | 0 | 0 | |
| B9[*] | other service facilities | 0.23 | 350 | 2,308 | 0.64 | 84 | 48 | |
| M1[*] | class I industrial | 0.60 | 20,914 | 336 | 0.81 | 41 | 30 | |
| M2[*] | class II industrial | 0.40 | 146,229 | 32,592 | 0.52 | 1,692 | 1,059 | |
| M3[*] | class III industrial | 0.30 | 39,284 | 10,132 | 0.55 | 418 | 238 | |
| W1[*] | Warehouse | 0.50 | 7,500 | 7,904 | 0.85 | 836 | 607 | |
| E2[**] | agriculture and forest | 0.004 | 1,398,392 | 3,943,224 | 0.65 | 637 | 637 | 5506 (agriculture) |
| R1[*] | class I residential | 0.35 | 3,312 | 0 | 0.00 | 0 | 0 | |
| R2[*] | class II residential | 0.25 | 104,612 | 232,924 | 0.53 | 7,667 | 9,435 | 25080 (family property) |
| R3[*] | class III residential | 0.20 | 35,894 | 78,672 | 0.70 | 2,757 | 3,189 | |
| H14[*] | village construction | 0.13 | 425,821 | 176,868 | 0.78 | 4,508 | 5,189 | |
| S43[*] | community parking | 0.50 | 63,236 | 66,468 | 0.75 | 6,192 | 7,267 | |
| H3[**] | regional public facilities | 0.15 | 350 | 0 | 0.00 | 0 | 0 | |
| H41[**] | military sites | 0.30 | 630 | 0 | 0.00 | 0 | 0 | |
| H42[**] | security | 0.08 | 468 | 0 | 0.00 | 0 | 0 | |
| S1[**] | urban road | 0.12 | 1,055,377 | 931,240 | 0.07 | 3,048 | 3,141 | |
| S3[**] | transportation hub | 0.28 | 5,560 | 4,592 | 0.08 | 25 | 26 | |
| S41[**] | public transport station | 0.40 | 0.8 | 0 | 0.00 | 0 | 0 | |
| S42[**] | social parking | 0.60 | 2,107 | 0 | 0.00 | 0 | 0 | |
| S9[**] | other transportation facilities | 0.15 | 285 | 4 | 0.44 | 0.07 | 0.07 | |
| U11[**] | water supply | 0.27 | 857 | 0 | 0.00 | 0 | 0 | |
| U12[**] | power supply | 0.30 | 931 | 8 | 0.06 | 0.04 | 0.04 | 3448 (infrastructure) |
| U13[**] | gas supply | 0.40 | 108 | 0 | 0.00 | 0 | 0 | |
| U15[**] | communication facilities | 0.80 | 3,294 | 0 | 0.00 | 0 | 0 | |
| U16[**] | radio and television facilities | 0.80 | 809 | 0 | 0.00 | 0 | 0 | |
| U2[**] | environmental facilities | 0.50 | 141 | 0 | 0.00 | 0 | 0 | |
| U21[**] | drainage facilities | 0.45 | 757 | 3,352 | 0.25 | 94 | 96 | |
| U22[**] | sanitation facilities | 0.15 | 197 | 544 | 0.17 | 3.4 | 3.5 | |
| U31[**] | fire control facilities | 0.30 | 727 | 0 | 0.00 | 0 | 0 | |
| U9[**] | other public facilities | 0.15 | 21 | 0 | 0.00 | 0 | 0 | |
| G2[**] | protection greenbelt | 0.01 | 713 | 0 | 0.00 | 0 | 0 | |

| G3** | square land | 0.08 | 1179 | 28,192 | 0.41 | 230 | 234 | |
| G1** | park green space | 0.01 | 48,265 | 1,608,096 | 0.55 | 2,212 | 2,212 | |
| B3* | recreational and sports facilities | 0.38 | 18,235 | 161,216 | 0.88 | 13,541 | 13,541 | |

* The estimated exposure value is the indoor property value, ** the estimated value is the value or cost per unit area.

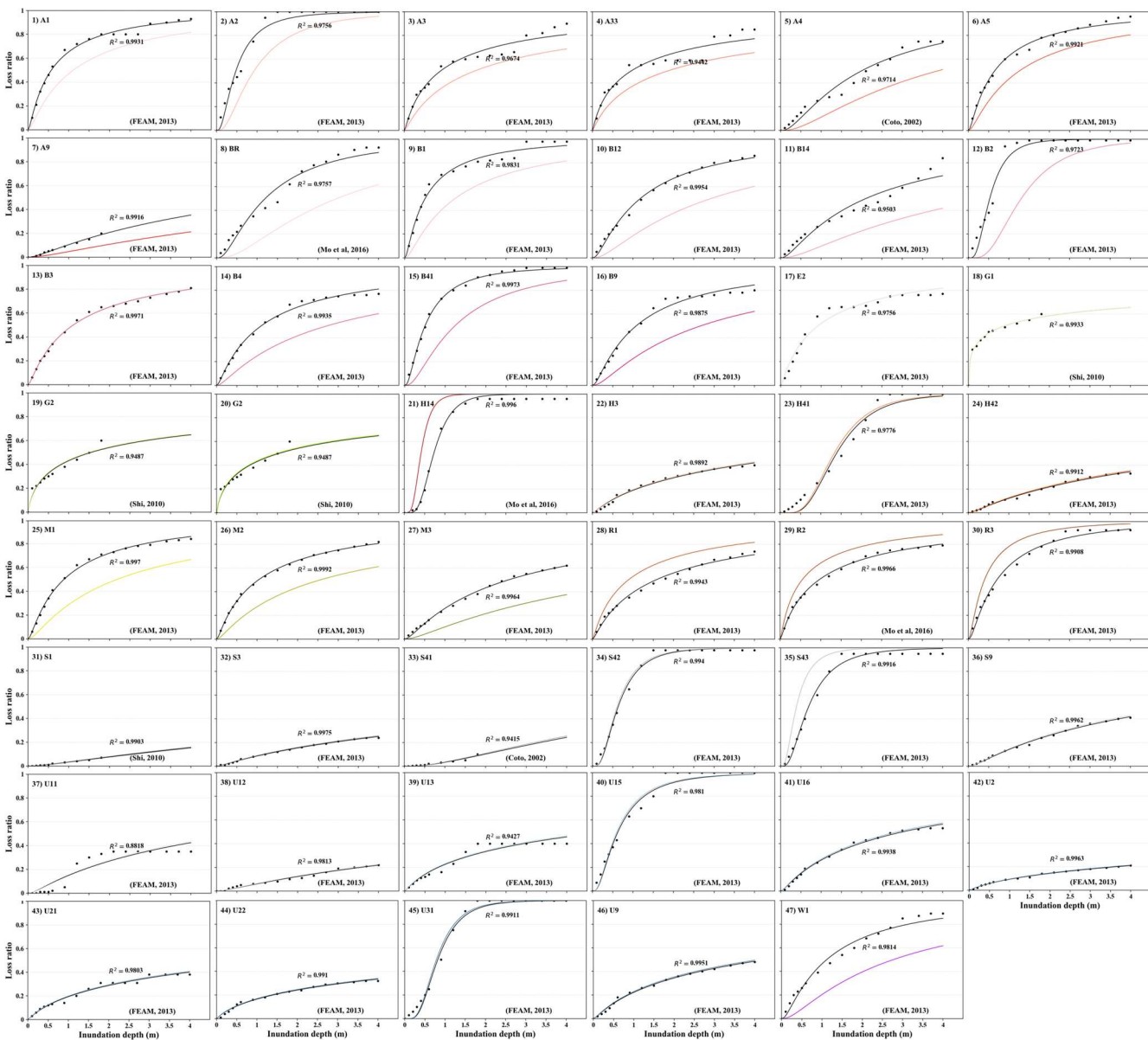

**Figure 6. Fitting and optimization of vulnerability curves of landuse in Liandu district (The black dots refer to the inundation depth and loss ratio based on the reference, the black curve is the fitting result of the black dots by the lognormal CDF, and the color curve is the optimized result based on the disaster loss reporting).**

The loss ratios of residential, industrial, commercial, warehousing, public management, and public service land are very high, mainly due to indoor properties being soaked or washed away by floods. As the inundation depth increases, the loss ratio increases rapidly. When the depth is higher than 3 m, the increase of loss ratio is not obvious.

The direct impact of floods on public facilities and roads is relatively small, and the loss ratio is generally low. However, the indirect loss caused by the suspension of roads, communications, and electricity is relatively large but is not calculated in this study. Green space and square land are less affected by floods, and the loss ratio is relatively low.

The scale parameters of the vulnerability curve of the landuse type corresponding to each statistical indicator in disaster report selected the same optimization coefficient, and the coefficient was obtained by solving the nonlinear equation. The vulnerability curve of family property is stretched, that of infrastructure is stretched slightly, and those of public welfare facilities, industry, and commerce shrink. This comparison shows that the simulated loss after optimization is consistent with the disaster loss reporting, and the optimization effect is good (Table 2).

## 4.4 Distributions of the loss ratio and loss value

Based on the landuse type and value data, the inundation depth distribution, and the optimized vulnerability functions of the landuse, the distribution of loss ratio (Figure 7a) and loss value (Figure 7b) of Liandu district were estimated by the method in step 3 of Section 3.3.

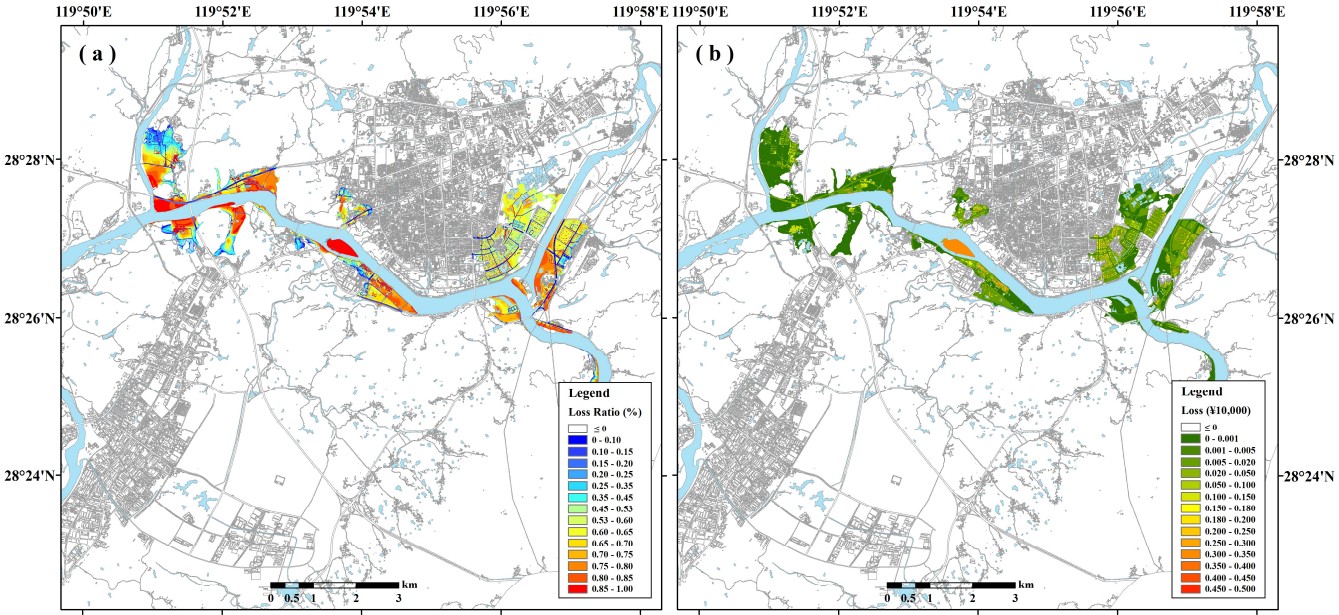

**Figure 7. Spatial distribution of the flood loss ratio (a) and loss values (b) in the central urban area of Liandu district in 2014.**

Due to the high inundation depth at the river confluence and both sides of the river as well as the wide distribution of agricultural land in the area, the mean loss ratio of these areas is approximately 0.63. The inundation depth of the island is more than 7 m, and there are entertainment facilities on it, so the loss ratio is more than 0.6. In addition, the inundation depth

on both sides of the Hao River is more than 1 m. The west side of the river is mainly residential land, and the east side is the Lishui railway station, so the indoor property loss ratio is high. However, traffic land is less directly affected by floods, so the loss ratio is less than 0.2 (Figure 7a).

The loss value of a flood disaster is affected by the unit value and loss ratio of the landuse. Therefore, the distribution characteristics of the loss ratio and loss value are different. The loss ratio of this flood is relatively high, but due to the wide distribution of agricultural land in the submerged area, the loss value per unit area is low. The loss ratio and unit area value of recreational and sports facilities, residential, industrial, commercial, public management, and public service facilities are high, so the loss values are also high. The loss ratio of traffic facilities is very low, yet the loss value is relatively high due to the high construction cost (Figure 7b).

Liandu district is surrounded by high mountains, and the overland flow, affected by the topography, was formed by rainfall flowing into the Da River through the rivers in the north. The occurrences of heavy rainfall in the middle and upper reaches of the Oujiang River are similar, causing the whole basin to experience the flood peak period at the same time, which is not conducive to flood diversion efforts. To relieve the flood pressure on the basin, the Jinshuitan Reservoir was opened twice. At the same time, rapid urbanization has brought about great changes in the morphology of river channels and waters. These factors caused the river's water level to rise rapidly, overtopping the flood dyke and submerging low-lying areas along the riverbed. In addition, due to the high population density and highly concentrated economy in the northern area of Liandu district, serious economic losses will inevitably occur when floods exceed the fortification standard of dyke.

## 5 Conclusion and discussion

The research and verification of the refined assessment of single-flood disaster loss was carried out by utilizing the refined types and values of landuse and quantitative vulnerability curves, and the main conclusions are as follows.

The 47 types of landuse data obtained by the fusion of vector and raster data overcome the limitations of the original data. This procedure not only refines the data into detailed urban landuse types but also has a high spatial resolution. In addition, in the absence of a comprehensive and detailed survey of exposure data, the unit area costs or asset values obtained through the collection of written materials and expert questionnaires can reflect the overall distribution of the value of the exposure data to a certain extent. The types and values of landuse developed by the above methods provide precise and reasonable exposure data for the refined assessment of disaster losses.

Based on lognormal cumulative distribution function fitting and scale parameter optimization, the vulnerability curves of 47 landuse types accurately reflect the characteristics of landuse loss ratio varying with the inundation depth in Liandu district and provide a reliable depth damage function curve for refined loss assessment. Among these landuse types, residential, industrial, commercial, public management, and storage land are seriously affected by flooding. Other landuses are relatively less affected, and the loss ratio increases slowly. In the absence of a large amount of post-disaster field survey data, the method proposed in this study can be used to construct vulnerability curves in accordance with the regional situation.

A refined assessment model of the direct economic loss of a single flood disaster is developed in accordance with the regional characteristics based on the refined research and verification of each link in the disaster loss assessment. The estimated spatial distributions of the loss ratio and loss value of the flood disaster accurately reflect the intensity and spatial pattern of disaster loss, which is conducive to the government's rational deployment of rescue forces and effective emergency assistance, especially in areas with severe disasters to increase rescue forces and evacuate the people in time to reduce losses to a certain extent. Expect for park green space, recreational and sports facilities, and agricultural and forestry land, the total loss of other landuse types after optimization is 322.6 million RMB, which is 510,000 RMB higher than the loss report data. The error between the two is relatively small, indicating that the loss assessment model of this study can be effectively applied to this area.

The loss assessment model developed in this study can be used to estimate flood disaster losses under various climate change and socio-economic scenarios, providing a basis for flood risk assessment and management in small- and medium-sized cities. This, in turn, will help the government formulate reasonable climate adaptation policies and sponge city planning (MHURD, 2014).

In this study, only the inundation and disaster loss reporting of one precipitation event are collected, thereby affecting, to a certain extent, the optimization results of the vulnerability curves. In the future, the data of several precipitation events will be collected to better calibrate the vulnerability curves and render the final optimized vulnerability curves more suitable for the region. Furthermore, the flood disaster caused serious direct economic losses to Liandu district. The disaster also stimulated a large amount of disaster relief investment, even as it disrupted public services and traffic, caused production losses for companies outside the flooded area, reduced agricultural production, and affected related industries (Jonkman et al., 2008). Therefore, it is necessary to carry out further research regarding the refined assessment of the indirect economic losses of flood disasters.

***Code and data availability.*** The data used in the study are available at https://github.com/Haixia-Zhang/Flood-loss-assessment.git

***Author contributions.*** FWH and ZHX conceived the research framework and developed the methodology. ZHX was responsible for the code compilation, data analysis, graphic visualization, and first draft writing. FWH managed the implementation of research activities and revised the manuscript following reviewers' suggestions. ZH and YL participated in the data collection and data curation of this study. All authors discussed the results and contributed to the final version of the paper.

***Competing interests.*** The authors declare that they have no conflict of interest.

***Special issue statement.*** This article is part of the special issue "Advances in flood forecasting and early warning". It is not associated with a conference.

*Acknowledgements.* This work was mainly supported by the National Key Research and Development Program of China (grant nos. 2018YFC1508803 and 2017YFA0604903), and jointly supported by the Key Special Project for Introduced Talents Team of Southern Marine Science and Engineering Guangdong Laboratory (Guangzhou) (No. GML2019ZD0601) and Deutsche Gesellschaft für Internationale Zusammenarbeit (GIZ) GmbH. The authors would like to acknowledge Kang Ying and Lin Song of the *Zhejiang Design Institute of Water Conservancy & Hydro-Electric Power* for providing the flood inundation data in Liandu district.

*Financial support.* This research has been supported by the National Key Research and Development Program of China (grant nos. 2018YFC1508803 and 2017YFA0604903), the Key Special Project for Introduced Talents Team of Southern Marine Science and Engineering Guangdong Laboratory (Guangzhou) (grant no. GML2019ZD0601), and Deutsche Gesellschaft für Internationale Zusammenarbeit (GIZ) GmbH.

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
