# Peer review of "Assessment of Direct Economic Losses of Flood Disasters Based on Spatial Valuation of Landuse and Quantification of Vulnerabilities: A Case Study on the 2014 Flood in Lishui city of China"

_Natural Hazards and Earth System Sciences, 2021_

## Author Comment (AC1)

We sincerely thank Jean-Paul Pinelli for the valuable feedback, which we have used to improve the quality of our manuscript. The referee comments are provided below in **bold font,** and specific concerns have been numbered. Our responses are given in normal font, and changes/additions to the manuscript are given in blue text.

**General comments**

1. **This is a very interesting paper in a much-needed area. The paper proposes a methodology to evaluate inland flood losses for specific events, which potentially could be used for flood disaster management and prevention.**

   **Response:** Thanks for your comments. We greatly appreciate your kind help in the reviewing the manuscript.

**Specific comments**

1. **In the conclusion, the authors state that "The estimated spatial distributions of the loss ratio and loss value of the flood disaster accurately reflect the spatial pattern of disaster loss and provide scientific guidance for disaster prevention, mitigation, and emergency rescue". This is certainly true, but the paper would benefit from a more detailed explanation on how the work could actually be used for guidance. May be a whole section on this topic would be welcome.**

   **Response:** Thanks for your comments. This research focuses on the construction and verification of the refined assessment model of the direct economic loss, so we did not set up a section to introduce the practical application of the loss simulation results, but we have added an introduction to the practical application of the simulated loss results in Section 5.

   **Line 323-327:** The estimated spatial distributions of the loss ratio and loss value of the flood disaster accurately reflect the intensity and spatial pattern of disaster loss, which is conducive to the government's rational deployment of rescue forces and effective emergency assistance, especially in areas with severe disasters to increase rescue forces and evacuate the people in time to reduce losses to a certain extent.

2. **Also, the procedure for the derivation of the vulnerability curves is not clear, and should be improved and consolidated.**

   **Response:** Thanks for your comments. In the revised manuscript, we have reorganized the process of fitting and optimizing the vulnerability functions, and described in detail the steps that were not clear in the original manuscript.

   **Line 186-225:** Therefore, based on the existing vulnerability curves in many countries and regions (Coto, 2002; FEMA, 2013; Mo and Fang, 2016; Shi, 2010), the fitting and optimization steps of vulnerability function are as follows.
   1) According to the existing depth-damage database, the relationship between inundation depth and loss ratio of each land use in Liandu district was constructed. Since the HAZUS-

FLOOD had a complete building occupancy class, this study mainly referred to it. First, we constructed the comparison table between the building occupancy class in HAZUS-FLOOD and the land use type in Liandu district. Second, based on the inundation depth in Liandu district and the water depth in HAZUS-FLOOD, the range of inundation depth in the depth-damage function was set and the unit of water depth was converted to meters. Finally, the HAZUS-FLOOD was summarized according to the building occupation class, and the average loss ratio of all samples for each building occupancy class under each inundation depth was calculated, which was used as the reference of the loss ratio of corresponding land use types under the same inundation depth in Liandu district. If there was no similar building occupancy type in HAZUS-FLOOD, other databases were referenced (Figure 6).

2) The appropriate function was selected to fit the curve of inundation depth and loss ratio for each land use type constructed in step 1. In the previous study, the vulnerability curve can be fitted by a polynomial, a power function (Büchele et al., 2006), or logistic regression (Cao et al., 2016), and it can also be smoothed by nonparametric forms such as the kernel density (Merz et al., 2004). In this paper, the lognormal cumulative distribution function (Limpert et al., 2001) was selected to fit the vulnerability curve. The formula is as follows:

$$y = F(x, scale, shape) = F(x|\mu, \sigma) = \frac{1}{\sigma\sqrt{2\pi}} \int_0^x \frac{1}{t} e^{\frac{-(\log t - \mu)^2}{2\sigma^2}} dt, \quad x > 0 \qquad (1)$$

where $y$ is the loss ratio, $x$ is the inundation depth, $\sigma$ and $\mu$ are the standard deviation and mean of the $\log x$, respectively. For lognormal cumulative distribution function $F$, the shape parameter $shape = \sigma$, which affects the shape of the distribution, and scale parameter $scale = e^\mu$, which affects the stretching and shrinking of the distribution.

3) Based on the vulnerability function fitted in step 2, the loss ratio and loss value of land use in Liandu district were estimated. First, the two raster layers of land use type and inundation depth were overlaid. Then, the loss ratio of each grid was calculated based on the vulnerability function. If the inundation depth of the grid was 0 m, the loss ratio was also 0. Otherwise, the corresponding vulnerability function was searched according to the land use type of the grid, and its loss ratio was calculated based on the inundation depth of the grid. Finally, the loss value of each grid was calculated by Eq. 2.

$$L = DR * V \qquad (2)$$

where $L$ is the loss value of the land use, $DR$ is the loss ratio of the land use, and $V$ is the value of the land use.

4) The vulnerability function was optimized by disaster loss reporting data. First, the mapping table between the statistical indicators of disaster loss reporting and land use types in Liandu district was constructed (Table 2). Second, the simulated total loss of land use in Liandu district corresponding to the loss reporting data of indicator $k$ was calculated by Eq. 3. Then, the non-linear equation was established with the minimum error between the disaster reporting loss and the simulated total loss as the objective function, and the scaling factor $a_k$ was solved by the least square method.

$$TL_k = \sum_{i=1}^{n_k} \sum_{j=1}^{m_{ik}} L_{ij} = \sum_{i=1}^{n_k} \sum_{j=1}^{m_{ik}} F(x_{ij}, shape_i, a_k * scale_i) * value_i \quad (k = 1,2,...,5) \quad (3)$$

where $TL_k$ is the simulated total loss, $n_k$ is the number of land use types corresponding to the disaster loss reporting indicator $k$, $m_{ik}$ is the number of grids of the land use type

$i$ corresponding to the indicator $k$, $L_{ij}$ and $F(x_{ij}, shape_i, a_k * scale_i)$ are the loss value and loss ratio of the grid $j$ of the land use type $i$, respectively. $value_i$ is the asset value or cost per unit area of land use type $i$, $x_{ij}$ is the inundation depth of the grid $j$ of the land use type $i$, $shape_i$ and $scale_i$ are the parameter of the vulnerability function of the land use type $i$, $a_k$ is the scaling factor of the indicator $k$, and the initial value is 1.

**Technical and editorial comments**

1. **Overall, the paper would benefit if the authors would try to reduce the use of the passive voice.**

   **Response:** We have carefully read the full manuscript and modified some passive voice to active voice according to the comment.

2. **Line 16: "values of disaster-bearing bodies." Not sure what is meant. I suggest you use another expression.**

   **Response:** This suggestion has been adopted, and we have corrected "values of disaster-bearing bodies" to "values of land use". In addition, we have replaced all the phrases "disaster-bearing bodies" in the manuscript with appropriate words.

   **Line 14-16:** Next, the urban land use status map and high-resolution land use classifications based on remote sensing data were fused and combined with expert questionnaire surveys, thereby providing the 47 types and values of land use.

   **Line 48-50:** and other flood-prone countries have carried out a large number of loss assessment studies using different classification systems of exposure data and then used the existing loss database and post-disaster investigation data to establish local flood vulnerability functions.

   **Line 55-56:** Second, the effect and accuracy of the assessment are affected by the scale of the exposure data.

   **Line 57-58:** However, mesoscale exposure data mainly refer to land use obtained through remote sensing (RS) interpretation (Merz et al., 2010).

   **Line 314-315:** The types and values of land use constructed by the above methods provide precise and reasonable exposure data for the refined assessment of disaster losses.

   **Line 316-319:** Based on lognormal cumulative distribution function fitting and scale parameter optimization, the vulnerability curves of 47 land use types accurately reflect the characteristics of land use loss ratio varying with the inundation depth in Liandu district and provide a reliable depth damage function curve for refined loss assessment. Among these land use types, residential, industrial, commercial, public management, and storage land are seriously affected by flooding.

3. **Line 29: not sure what a "sponge" city is.**

**Response:** Sponge city refers to a city that can be as flexible as a sponge in adapting to environmental changes and natural disasters. It absorbs, stores, infiltrates and purifies water when it rains, and releases and reuses the stored water when necessary. It is a sustainable urban development model, which can effectively alleviate urban waterlogging, reduce runoff pollution, save water resources and improve ecological environment. According to your suggestion, we add a brief description of sponge city in the revised manuscript.

**Line 32-34:** Among them, sponge city refers to a city that can be as flexible as a sponge in adapting to environmental changes and natural disasters. It absorbs, stores, infiltrates and purifies water when it rains, and releases and reuses the stored water when necessary (Yu et al., 2015).

*Yu, K., Li, D., Yuan, H., Fu, W., Qiao, Q. and Wang, S.: "Sponge City": theory and practice, City Plan. Rev., 39(6), 26–36, doi:10.11819/cpr20150605a, 2015.*

4. **Line 40: I suggest using more recent references for the USA.**

   **Response:** Thanks for your suggestion, and we have modified the references in the revised manuscript.

   **Line 45-50:** The United States(Custer and Nishijima, 2015; USACE, 2006), the United Kingdom (Stephenson and D'Ayala, 2013), Japan (Dutta et al., 2003), Canada (NRC, 2017), Australia (Hasanzadeh Nafari et al., 2016b, 2016a; Wehner et al., 2017), Italy (Amadio et al., 2016), China (Li et al., 2012; Penning-Rowsell et al., 2013), and other flood-prone countries have carried out a large number of loss assessment studies using different classification systems of exposure data and then used the existing loss database and post-disaster investigation data to establish local flood vulnerability functions.

   *Custer, R. and Nishijima, K.: Flood vulnerability assessment of residential buildings by explicit damage process modelling, Springer Netherlands., 2015.*

   *USACE (United States Army Corps of Engineers): Depth-damage relationships for structures, contents, and vehicles and content-to-structure value ratios (CSVR) in support of the Donaldsonville to the Gulf, Louisiana, feasibility study, New Orleans District, Louisiana., 2006.*

5. **Line 54: I suggest that you add as a reference, "J.-P. Pinelli, Josemar Da Cruz, K. Gurley, A. Paleo-Torres, M. Baradaranshoraka, S. Cocke & D.-W. Shin, " Data management for the development, validation, calibration, and operation of a hurricane vulnerability model," International Journal of Disaster Risk Science, November 2020, DOI 10.1007/s13753-020-00316-4"**

   **Response:** This suggestion has been adopted, and this article has been added as a reference for the uncertainty of flood loss assessment.

   **Line 59-60:** These problems lead to high uncertainties and disparities in flood loss assessments (Gerl et al., 2016; Pinelli et al., 2020).

*Gerl, T., Kreibich, H., Franco, G., Marechal, D. and Schröter, K.: A review of flood loss models as basis for harmonization and benchmarking, PLoS One, 11(7), doi:10.1371/journal.pone.0159791, 2016.*

*Pinelli, J. P., Da Cruz, J., Gurley, K., Paleo-Torres, A. S., Baradaranshoraka, M., Cocke, S. and Shin, D.: Uncertainty reduction through data management in the development, validation, calibration, and operation of a hurricane vulnerability model, Int. J. Disaster Risk Sci., 11(6), 790–806, doi:10.1007/s13753-020-00316-4, 2020.*

6. **Line 76: is this correct: the district is in the city? Or should it be the reverse?**

   **Response:** Thanks for your comment. However, the location of district and city is not reversed in this study. In China, the administrative divisions are divided into five levels: province, prefecture, county, township and village. Among them, the prefecture-level regions include 17 prefectures, 30 autonomous prefectures, 283 prefecture level cities and 3 leagues. County-level regions include 1464 counties, 117 autonomous counties, 374 county-level cities, 852 districts, 49 banners, 3 autonomous banners, a forest area and 2 special districts. Lishui city and Liandu district in this study belong to prefecture-level region and county-level region respectively, so the district is in the city. To make it easier to understand, we have used the appropriate expressions in the revised manuscript.

   **Line 80-81:** Liandu is a district under the jurisdiction of Lishui city, with a relatively concentrated population and socioeconomic status.

7. **Line139: Not if the expression remote sensing is the correct one. Remote sensing refers to the acquisition of data through sensors at a distance, hence remote. From your description this is not the case. It seems you are describing process of digitization.**

   **Response:** Thanks for your comments. In this manuscript, "remote sensing classification" actually refers to "land use classification results based on remote sensing images", and we have modified all the phrases "remote sensing classification" in the revised manuscript.

   **Line 14-16:** Next, the urban land use status map and high-resolution land use classifications based on remote sensing data were fused and combined with expert questionnaire surveys, thereby providing the 47 types and values of land use.

   **Line 152-153:** The distribution of land use types was obtained through the fusion of current land use data in Lishui city with high resolution land use classification results based on remote sensing data.

   **Line 155-157:** The land use classification results were derived from the Gaode map with a resolution of 2.3870768 m, including 5 categories: transportation, grassland, waters, agriculture and forest, and buildings.

   **Line 163-164:** Second, we extracted the land use type in the current land use data corresponding to the location of each building pixel in the land use classification result, and assigned the type to the corresponding building pixel.

   **Line 167-168:** The water, agriculture and forest, and road in the land use classification

results were assigned as water, agriculture and forest, and urban road, respectively.

**Line 170:** Whether there were un-reassigned pixels in the land use classification results.

**Line 231-232:** The fused land use type data effectively integrate the respective advantages of the current urban land use and land use classification results based on remote sensing images.

8. **Line 146: Not sure what is meant by "traversed". Please, clarify.**

   **Response:** Thanks for your comments. In the manuscript, "traversed" refers to the operation of building pixels in the land use classification result one by one, and now we have corrected the expression of this sentence.

   **Line 163-164:** Second, we extracted the land use type in the current land use data corresponding to the location of each building pixel in the land use classification result, and assigned the type to the corresponding building pixel.

9. **Line 170: you might want to reference and look into different reports from the US Corps of Engineers. I believe that HAZUS inland flood curves are actually derived from their work.**

   **Response:** Thanks for your comments. We have corrected the reference here to include the vulnerability curve in this study.

   **Line 186-188:** Therefore, based on the existing vulnerability curves in many countries and regions (Coto, 2002; FEMA, 2013; Mo and Fang, 2016; Shi, 2010), the fitting and optimization steps of vulnerability function are as follows.

   *Coto, E. B.: Flood hazard, vulnerability and risk assessment in the city of Turrialba, Costa Rica, International Institute for Geo-information Science and Earth Observation(ITC), Enschede, The Netherlands. [online] Available from: http://www.itc.nl/library/Papers/msc_2002/ereg/badilla_coto.pdf, 2002.*

   *FEMA (Federal Emergency Management Agency): Multi-hazard loss estimation methodology. HAZUS-MH flood model technical manual., 2013.*

   *Mo, W. and Fang, W.: Empirical vulnerability functions of building contents to flood based on post-typhoon (Fitow, 201323) questionnaire survey in Yuyao, Zhejiang, Trop. Geogr., 36(4), 633-641+657, doi:https://doi.org/10.13284/j.cnki.rddl.002828, 2016.*

   *Shi, Y.: Research on vulnerability assessment of cities on the disaster scenario: A case study of Shanghai city, East China Normal University, Shanghai, China., 2010.*

10. **Section 3.3 and line 221: If I understand correctly, you have data of accumulated loss as a function of land use. For each type of land use, you adjust the parameters of the equation of your vulnerability curve till you get the optimum fit (least error between model and data). In number 4) you mention the "scale parameters", but I am not clear what they are. Could you please clarify?**

**Response:** We have added the description of scale parameter and shape parameter of the lognormal cumulative distribution function in the revised manuscript. In this study, lognormal cumulative distribution function is used to fit the vulnerability curve, where shape parameter $shape = \sigma$, scale parameter $scale = e^{\mu}$. The shape parameter affects the shape of the function distribution, and the scale parameter affects the stretching and shrinking of function distribution. The change trends of inundation depth and loss ratio in different regions are generally the same. Therefore, the accumulated loss of different land use types is used to optimize the scale parameter of its vulnerability function to obtain the optimized vulnerability curve.

**Line 205-206:** For lognormal cumulative distribution function $F$, the shape parameter $shape = \sigma$, which affects the shape of the distribution, and scale parameter $scale = e^{\mu}$, which affects the stretching and shrinking of the distribution.

11. **Now I see your figure 6, but you have a reference there to FEAM 2013, which I cannot find in your list of references. Also, in the figure caption you write that the color curve is the optimized result, but the graphs seem to indicate the reverse, with the black being the best fit to the data. Am I missing something? Am I correct in interpreting that the black dots in your figures are data points from the references, to which you initially fit you VC, and then you adjust some kind of scale parameter to get a new optimized fit to your own data for the Liandu event? This is somewhat confusing and need to be better explained and clarified.**

**Response:** The black dots in Fig.6 refer to the inundation depth and loss ratio constructed according to the corresponding reference. The black curve is obtained by fitting the inundation depth and the loss ratio (black dots) through the lognormal cumulative distribution function, which is the vulnerability curve. Then, based on the disaster loss reporting, the scale parameters of the lognormal cumulative distribution function (black curve) of different land use are optimized to obtain the optimized vulnerability curve (color curve), and the curve uses the same color as the land use type in Fig.5a.

**Line 391-392:** *FEMA (Federal Emergency Management Agency): Multi-hazard loss estimation methodology. HAZUS-MH flood model technical manual., 2013.*

**Line 267-269:** Figure 6. Fitting and optimization of vulnerability curves of land use in Liandu district (The black dots refer to the inundation depth and loss ratio based on the reference, the black curve is the fitting result of the black dots by the lognormal CDF, and the color curve is the optimized result based on the disaster loss reporting).

12. **Line 262: fortification should be replaced by mitigation. If you are not referring to mitigation, then, please clarify the meaning.**

**Response:** Thanks for your comment. The statement in the manuscript is incomplete. It refers to the fortification standard of the dyke, and we have corrected it.

**Line 305-306:** In addition, due to the high population density and highly concentrated economy in the northern area of Liandu district, serious economic losses will inevitably

occur when floods exceed the fortification standard of dyke.

**13. Line 269: "more reasonable" than what? The last sentence of that paragraph should be rewritten to clarify the meaning.**

**Response:** Thanks for your comments. The expression "more reasonable" in the manuscript is not rigorous, and we have corrected it in the revised manuscript. In addition, the last sentence has been rewritten to make its meaning clear.

**Line 311-314:** In addition, in the absence of a comprehensive and detailed survey of exposure data, the unit area costs or asset values obtained through the collection of written materials and expert questionnaires can reflect the overall distribution of the value of the exposure data to a certain extent.

**Line 314-315:** The types and values of land use constructed by the above methods provide precise and reasonable exposure data for the refined assessment of disaster losses.

**14. Line 286: I don't know what sponge city planning means.**

**Response:** Rapid urbanization in China has caused severe water shortage, waterlogging, and water pollution problems in recent years. To solve these issues, the Chinese government launched the sponge city construction planning on December 31, 2014 (MHURD, 2014). The primary goals of China's sponge city construction are: absorbing and reusing 70% - 90% of rainfall by applying the green infrastructure concept and using low impact development (LID) measures, eliminating water logging and preventing urban flooding, improving urban water quality, and minimizing the impact of urban development and construction on the eco-systems.

**Line 34-37:** In order to solve the problems of water shortage, waterlogging and water pollution caused by rapid urbanization, the Chinese government launched the sponge city construction plan on December 31, 2014 (MHURD, 2014), which can effectively alleviate urban waterlogging, reduce runoff pollution, save water resources and improve ecological environment.

*Ministry of Housing and Urban-Rural Development (MHURD): Technical guide for sponge cities-water system construction of low impact development. [online] Available from: http://jst.jl.gov.cn/csjs/wjxx/201412/P020141222565834965487.pdf, 2014.*

---

## Author Comment (AC2)

We sincerely appreciate the Referee 2 for the positive and constructive comments, which we have used to improve the quality of our manuscript. The referee comments are provided below in **bold font**, and specific concerns have been numbered. Our responses are given in normal font, and changes/additions to the manuscript are given in blue text.

**General comments**

1. The paper aims to build a refined assessment model of direct economic losses for flood disasters. Taking the "8.20" flood event hitting Linshui City in 2014 as an example, the loss ratio and loss value of this disaster event was calculated based on the land use data. The topic is interesting, and the paper is well structured. I would recommend following suggestions to improve the manuscript:

**Response:** We greatly appreciate your kind help in the reviewing the manuscript and all constructive comments. And we have revised the manuscript based on these comments and suggestions.

**Specific comments**

1. Line 14: please clarify what is the "flooding model".

**Response:** The "flooding model" in this sentence refer to "one-dimensional hydrodynamic model and geographic information system (GIS) analysis method", which we have modified in the revised manuscript.

Line 12-14: First, based on a field investigation, the inundation data simulated by the onedimensional hydrodynamic model and geographic information system (GIS) analysis method were verified.

2. Line 16-17: "Then, the existing vulnerability curve database was summarized, and the curves were calibrated by disaster loss reporting." This sentence does not make it clear the process of constructing and checking the vulnerability function. I suggest that you rewrite this sentence.

**Response:** This suggestion has been adopted, and we have rewritten this sentence to make it clear.

**Line 16-17:** Then, based on the previous depth-damage function, the vulnerability curves of 47 types of land use in Liandu district were fitted by the lognormal cumulative distribution function and optimized using disaster loss report data.

3. Line 19: "high-precision spatial information" should be replaced by "high spatial resolution".

Response: We have corrected the phrase in the revised manuscript.

Line 18-19: It is found that the constructed land use data has detailed types and value attributes as well as high resolution.

**4. Line 70: there is no legend in the light blue area in Figure 1 (b).**

**Response:** The light blue area in Figure 1b refers to the Oujiang River Basin. In order to distinguish it from the sub basin, we set the fill color of the Oujiang River Basin as no color.

Line 74-77:

**Figure 1.** Location of the study area. (a. The location of Oujiang River Basin in China. b. The distribution of hydrological stations, precipitation stations, rivers and sub basins in Oujiang River Basin, and the location of Liandu district in Oujiang River Basin. c. The terrain distribution and town boundary of Liandu district.)

**5. Section 2.3: how to verify the results of flood inundation simulation? How accurate is the simulation of flooding? There is no detailed description of these in this paper, please give a brief description.**

**Response:** Thanks for your comments. We have described in detail the process of hydrodynamic model construction and verification in the revised manuscript.

Line 111-128: The flood inundation was calculated from the Yuxi to Kaitan Reservoir hydraulic model constructed by Zhejiang Design Institute of Water Conservancy & Hydro-Electric Power. In this study, the middle and upper reaches of the Oujiang River Basin were generalized into a mathematical model, the unsteady flow partial differential equations of the Saint-Venant open channel were used to describe the flood evolution process, and the implicit difference method was used to transform it into a difference

equation. Newton iteration and Gaussian principal component elimination method were used to solve the problem time by time, so as to obtain the water level and discharge of each section (Kang and Chen, 2007). This model considered weirs, gates, water-blocking bridges, water exchange between intervals, and flood detention areas, etc., and can be applied to the quantitative analysis and calculation of the flood evolution of this section of the river.

In order to make the flood evolution calculation can better simulate the water depth of this basin, the measured river section and the flood in 2014 were selected for simulation calculation to verify the accuracy of the mathematical model of the flood evolution calculation, and determine the parameters of the model. The simulation result of the measured flood on August 20, 2014 was carried out. The upper boundary used the discharge process of each reservoir and the measured data from the hydrological station, and the lower boundary used the measured water level. Comparing the model result with the measured flood traces and the measured process at the hydrological station, the difference in water level between the two was 0 m - 0.09 m, which shows that the model parameters were reasonable. The flood volume was calculated based on the simulated water level and the elevation of the embankment (Table 1). Based on the overflow volume and topographic data, the submerged area and inundation depth were estimated by GIS tools. Through measured flood traces, field surveys and aerial photograph, it was found that the simulated submergence results can well reflect the actual flood.

6. Line 140: it is recommended that you quote Table 2 in "47 categories" to indicate the specific classification.

Response: We have added a quote to Table 2 at "47 categories" in the revised manuscript.

Line 153-155: The former data came from the urban and rural space development current status map of the Natural Resources and Planning Bureau in 2013, which was divided into 47 categories (Table 2) according to the *Code for classification of urban land use and planning standards of development land* (MHURD, 2011).

**7. Line 141: you have a reference for GB 50137-2011 here, but it is not in your reference list. Please add it.**

**Response:** Thanks for your comment, and we have added the reference in revised manuscript.

Line 153-155: The former data came from the urban and rural space development current status map of the Natural Resources and Planning Bureau in 2013, which was divided into 47 categories (Table 2) according to the *Code for classification of urban land use and planning standards of development land* (MHURD, 2011).

Ministry of Housing and Urban-Rural Development (MHURD): Code for classification of urban land use and planning standards of development land, China., 2011.

8. Section 3.3: the steps for fitting and optimizing the vulnerability function of land use are not clear, and it is recommended to describe them in detail.

**Response:** In the revised manuscript, we have reorganized the process of fitting and optimizing the vulnerability functions of land use, and gave a detailed description of the steps that were not clear in the original manuscript.

**Line 186-225:** Therefore, based on the existing vulnerability curves in many countries and regions (Coto, 2002; FEMA, 2013; Mo and Fang, 2016; Shi, 2010), the fitting and optimization steps of vulnerability function are as follows.

1) According to the existing depth-damage database, the relationship between inundation depth and loss ratio of each land use in Liandu district was constructed. Since the HAZUS-FLOOD had a complete building occupancy class, this study mainly referred to it. First, we constructed the comparison table between the building occupancy class in HAZUS-FLOOD and the land use type in Liandu district. Second, based on the inundation depth in Liandu district and the water depth in HAZUS-FLOOD, the range of inundation depth in the depth-damage function was set and the unit of water depth was converted to meters. Finally, the HAZUS-FLOOD was summarized according to the building occupation class, and the average loss ratio of all samples for each building occupancy class under each inundation depth was calculated, which was used as the reference of the loss ratio of corresponding land use types under the same inundation depth in Liandu district. If there was no similar building occupancy type in HAZUS-FLOOD, other databases were referenced (Figure 2).

2) The appropriate function was selected to fit the curve of inundation depth and loss ratio for each land use type constructed in step 1. In the previous study, the vulnerability curve can be fitted by a polynomial, a power function (Büchele et al., 2006), or logistic regression (Cao et al., 2016), and it can also be smoothed by nonparametric forms such as the kernel density (Merz et al., 2004). In this paper, the lognormal cumulative distribution function (Limpert et al., 2001) was selected to fit the vulnerability curve. The formula is as follows:

$$y = F(x, scale, shape) = F(x|\mu, \sigma) = \frac{1}{\sigma\sqrt{2\pi}} \int_0^x \frac{1}{t} e^{\frac{-(\log t - \mu)^2}{2\sigma^2}} dt, \quad x > 0$$
(1)

where y is the loss ratio, x is the inundation depth,  $\sigma$  and  $\mu$  are the standard deviation and mean of the logx, respectively. For lognormal cumulative distribution function F, the shape parameter  $shape = \sigma$ , which affects the shape of the distribution, and scale parameter  $scale = e^{\mu}$ , which affects the stretching and shrinking of the distribution.

3) Based on the vulnerability function fitted in step 2, the loss ratio and loss value of land use in Liandu district were estimated. First, the two raster layers of land use type and inundation depth were overlaid. Then, the loss ratio of each grid was calculated based on the vulnerability function. If the inundation depth of the grid was 0 m, the loss ratio was also 0. Otherwise, the corresponding vulnerability function was searched according to the land use type of the grid, and its loss ratio was calculated based on the inundation depth of the grid. Finally, the loss value of each grid was calculated by Eq. 2.

$$L = DR * V$$

where L is the loss value of the land use, DR is the loss ratio of the land use, and V is the value of the land use.

(2)

4) The vulnerability function was optimized by disaster loss reporting data. First, the mapping table between the statistical indicators of disaster loss reporting and land use types in Liandu district was constructed (Table 2). Second, the simulated total loss of land

use in Liandu district corresponding to the loss reporting data of indicator k was calculated by Eq. 3. Then, the non-linear equation was established with the minimum error between the disaster reporting loss and the simulated total loss as the objective function, and the scaling factor  $a_k$  was solved by the least square method.

$$TL_{k} = \sum_{i=1}^{n_{k}} \sum_{j=1}^{m_{ik}} L_{ij} = \sum_{i=1}^{n_{k}} \sum_{j=1}^{m_{ik}} F(x_{ij}, shape_{i}, a_{k} * scale_{i}) * value_{i} \quad (k = 1, 2, ..., 5)$$
(3)

where  $TL_k$  is the simulated total loss,  $n_k$  is the number of land use types corresponding to the disaster loss reporting indicator k,  $m_{ik}$  is the number of grids of the land use type i corresponding to the indicator k,  $L_{ij}$  and  $F(x_{ij}, shape_i, a_k * scale_i)$  are the loss value and loss ratio of the grid j of the land use type i, respectively.  $value_i$  is the asset value or cost per unit area of land use type i,  $x_{ij}$  is the inundation depth of the grid j of the land use type i,  $shape_i$  and  $scale_i$  are the parameter of the vulnerability function of the land use type i,  $a_k$  is the scaling factor of the indicator k, and the initial value is 1.

**9. Line 173-174: please indicate which data source each type of land use refers to. You can quote the Figure 6.**

**Response:** Thanks for your suggestion, and we have added a quote to Figure 6 at the end of the sentence in the revised manuscript.

**Line 196-197:** If there was no similar building occupancy type in HAZUS-FLOOD, other databases were referenced (Figure 6).

**10. Line 186: "Based on the mapping relationships between the disaster statistical indicators and land use types". There are 41 statistical indicators for disaster loss reporting in Section 2.4. It is suggested to add this mapping table to determine which indicators are used in this study.**

**Response:** Thanks for your suggestion. We established the mapping relationship between the land use types of the Liandu district and the statistical indicators of the disaster report data in Table 2 in the original manuscript, but we did not add a quote to Table 2 in this sentence. Therefore, we add a quote in revised manuscript.

**Line 214-215:** First, the mapping table between the statistical indicators of disaster loss reporting and land use types in Liandu district was constructed (Table 2).

**11. Line 189: please clarify what is the "scale parameter".**

**Response:** We have added the description of the "scale parameter" in revised manuscript. The calculation formula of scale parameter of lognormal cumulative distribution function is  $scale = e^{\mu}$ , and the scale parameter affects the stretching and shrinking of function distribution.

Line 205-206: For lognormal cumulative distribution function *F*, the shape parameter  $shape = \sigma$ , which affects the shape of the distribution, and scale parameter  $scale = e^{\mu}$ , which affects the stretching and shrinking of the distribution.

**12. Line 193: "corresponding" should be replaced by "respective".**

Response: This word has been corrected in the revised manuscript.

Line 231-232: The fused land use type data effectively integrate the respective advantages of the current urban land use and land use classification results based on remote sensing images.

**13. Line 231: it is recommended to specify what the black dots in the Figure 6 refer to. In addition, Figure 6(42) should add the index and title of the horizontal axis.**

**Response:** The black dots in the Figure 6 refer to the inundation depth and loss ratio constructed according to the corresponding reference, and we have added the description in the title of Figure 6. In addition, we have added the tick labels and title of the horizontal axis in Figure 6(42).